# Comparison of the In Vivo Efficacy of Cuban (Raydel^®^) and Chinese (BOC Science) Policosanol in Alleviating Dyslipidemia and Inflammation via Safeguarding Major Organs and Reproductive Health in Hyperlipidemic Zebrafish: A Twelve-Week Consumption Study

**DOI:** 10.3390/ph17081103

**Published:** 2024-08-22

**Authors:** Kyung-Hyun Cho, Yunki Lee, Sang Hyuk Lee, Ji-Eun Kim, Ashutosh Bahuguna

**Affiliations:** Raydel Research Institute, Medical Innovation Complex, Daegu 41061, Republic of Korea

**Keywords:** apoptosis, dyslipidemia, fatty liver, interleukin (IL)-6, oxidative stress, policosanol, senescence

## Abstract

Policosanol is a blend of long-chain aliphatic alcohols (LCAAs) and is well-known for several health-beneficial activities; however, the functionality of policosanol varied substantially based on the composition of LCAAs. In this study, two distinct policosanols, Raydel^®^ (extracted from Cuban sugarcane wax) and BOC Sciences (extracted from Chinese sugarcane wax), were dietarily supplemented (0.1% *w*/*w*) for 12 weeks in hyperlipidemic zebrafish to examine their influence on the blood lipid profile and functionality of the liver, kidney, and reproductive organs. The results demonstrated a noteworthy impact of both policosanols on preventing high-cholesterol diet (HCD, 4% *w*/*w*)-induced dyslipidemia by decreasing total cholesterol (TC) and triglyceride (TG) levels in the plasma. However, compared to BOC Sciences, the Raydel^®^ policosanol exhibited a significantly (*p* < 0.05) higher efficacy in reducing HCD-induced TC and TG levels. A substantial effect was observed exclusively with the Raydel^®^ policosanol in mitigating HCD-impaired low-density-lipoprotein cholesterol (LDL-C) and high-density-lipoprotein cholesterol (HDL-C) levels. Hepatic histology and immunohistochemistry (IHC) analysis revealed the higher efficacy of Raydel^®^ policosanol over BOC Sciences policosanol to prevent HCD-provoked fatty liver changes, cellular senescence, oxidative stress, and interleukin (IL)-6 production. Consistently, a significantly higher effect of Raydel^®^ over BOC Sciences policosanol was observed on the protection of kidney, testis, and ovary morphology hampered by HCD consumption. In addition, Raydel^®^ policosanol exhibited a notably stronger effect (~2-fold, *p* < 0.05) on the egg-laying ability of the zebrafish compared to policosanol from BOC Sciences. Furthermore, Raydel^®^ policosanol plays a crucial role in improving embryo viability and mitigating developmental defects caused by the intake of an HCD. Conclusively, Raydel^®^ policosanol displayed a substantially higher efficacy over BOC Sciences policosanol to revert HCD-induced dyslipidemia, the functionality of vital organs, and the reproductive health of zebrafish.

## 1. Introduction

Policosanol is a typical blend of long-chain aliphatic alcohols (LCAAs) with a general CH_3_-(CH_2_)n-CH_3_-OH framework (where “n” varies from 22–32) and can be extracted from distinct sources such as maize, rice bran, wheat germ, sugar cane, and bee wax [1,2,3]. However, in 1993, policosanol was first extracted from Cuban sugarcane with a characteristic amalgamation of eight LCAAs viz C_24_H_49_OH (tetracosanol), C_26_H_53_OH (hexacosanol), C_27_H_55_OH (heptacosanol), C_28_H_57_OH (octacosanol), C_29_H_59_OH (nonacosanol), C_30_H_61_OH (triacontanol), C_32_H_65_OH (dotriacontanol), and C_34_H_69_OH (tetratriacontanol) [4,5].

The distinct functionality of policosanol has been logged by numerous in vitro and in vivo studies as reported by our [5,6] and other research groups [1]. However, the most noteworthy effect of policosanol was the reported ability to alleviate metabolic disorders including dyslipidemia [7] and hypertension [8]. A wide array of preclinical and clinical studies establishes an impactful role of policosanol in reducing total cholesterol (TC), triglycerides (TGs), and low-density-lipoprotein cholesterol (LDL-C) [1,9,10]. Besides the substantial effect of policosanol in inhibiting LDL oxidation [5], the elevation of high-density-lipoprotein cholesterol (HDL-C) levels and the functionality enhancement of the HDL-associated antioxidant enzyme paraoxonase (PON)-1 have been documented [11]. In some clinical studies, Cuban policosanol displayed a comparable effect with classical statins such as lovastatin [12,13,14], atorvastatin [15], simvastatin [16,17], pravastatin [18,19], acipimox [20], and probucol [21] in reducing elevated total cholesterol. In a clinical study with patients with type II diabetes and hypercholesterolemia, consumption of Cuban policosanol (10 mg/day) showed a better effect in reducing the LDL/HDL ratio and elevated HDL than lovastatin (10 mg/day) without exerting any side effects [12]. Furthermore, in combination with statins, policosanol effectively prevents statin-induced toxicity [22,23,24] toward liver and muscular impairment by diminishing coenzyme Q_10_ (CoQ_10_) [25,26,27,28]. A recent clinical study on obese participants established the substantial role of Cuban policosanol in conjunction with exercise in reducing body weight, improving blood lipid profile, reducing hypertension, and enhancing HDL functionality and antioxidant status without impending CoQ_10_ metabolism and liver damage [29].

Owing to the functionality and nontoxic nature of policosanol, more than 25 countries have approved policosanol consumption as a cholesterol-lowering nutraceutical and drug candidate [30]. Recently, a Scientific Statement from the American Heart Association enlisted policosanol as a complementary and alternative medicine for the management of heart failure [31]. Policosanol is gaining global acceptance, as evidenced by its increasing demand and trade, which is growing at a compound annual growth rate of 6.5% and is projected to reach USD ~600 million by 2032 [32]. Despite the several health-beneficial effects, there is wide variability and disparity in the effects exerted by various policosanols, which is primarily attributed to the LCAA composition of the policosanol. The composition of policosanol varies significantly based on the source material used, the extraction method, and the source material’s origin [1,33], which ultimately impact its functional properties.

The global market offers a variety of policosanol products derived from diverse source materials and geographical locations, yet comprehensive comparisons of their potency are scarce. To resolve this issue and provide a better choice to consumers, our previous studies entailed a comparative assessment of policosanol derived from different sources and locations, revealing considerable variation in their functionality [5,6,34]. In continuation, we have recently compared the in vitro efficacy of policosanol extracted from cane sugar wax originating from Cuba (Raydel^®^ policosanol, Havana, Cuba) and China (BOC Sciences policosanol, Shirley, NY, USA) and observed a substantial difference in their functionality toward the prevention of LDL oxidation, glycation of HDL, and carboxymethyllysine (CML)-induced toxicity in embryos and adult zebrafish [35].

Following the previous outcomes in the current investigation, the comparative efficacy of policosanol from Raydel^®^ and BOC Sciences was examined following 12 weeks of supplementation in hyperlipidemic zebrafish to determine their effect on maintaining the lipid profile and functionality of the liver, kidney, ovaries, and testis of hyperlipidemic zebrafish. In addition, the impact of policosanol from Raydel^®^ and BOC Sciences on egg-laying activity, the endurance of the laid eggs, and teratogenic effects was examined.

## 2. Results

### 2.1. Zebrafish Survivability and Change in Body Mass

The zebrafish showed no significant (*p* > 0.05) change in survivability across all the groups over 12 weeks. Across all the groups, the survivability of zebrafish remained almost constant, ranging from 92.8% (*n* = 52) to 100% (*n* = 56) [Appendix A].

The analysis of body weight demonstrated a progressive increase in body weight across all the groups over the 12-week consumption period (Figure 1A,B). After the 12 weeks, the HCD group exhibited the highest body weight (616.5 ± 35.1 mg), which was significantly greater (25.2%; *p* < 0.01) than the body weight observed in the ND control group, suggesting that an HCD has an effect on body weight enhancement. Conversely to the only HCD-fed group, the groups supplemented with Raydel^®^ (543.4 ± 29.6 mg) and BOC Sciences (576.2 ± 36.9 mg) policosanol displayed reduced body weights of 11.9% and 6.5%; however, this difference is statistically non-significant (*p* > 0.05).

### 2.2. Assessment of Plasma Lipid Profile

Figure 2 illustrates the effects of HCD feeding and policosanol consumption on the plasma lipid profile. A significantly higher (41.1%; *p* < 0.001) TC level was detected in the HCD-fed group in contrast to the ND control group. Consumption of Raydel^®^ and BOC Sciences policosanol led to significant reductions of 30.9% (*p* < 0.001) and 9.9% (*p* < 0.01) in TC levels, respectively, compared to the HCD-only group. Compared with the groups supplemented with BOC Sciences policosanol, TC levels were significantly lower (23.3%; *p* < 0.05) in the Raydel^®^ policosanol-enriched groups, suggesting the higher efficacy of Raydel^®^ policosanol in reducing HCD-induced TC levels.

Apart from TC, a significant effect of HCD consumption on the elevation of TG levels was detected, which was found to be 39.4% higher (*p* < 0.001) than basal TG levels detected in the ND control group (Figure 2). The consumption of BOC Sciences and Raydel^®^ policosanol substantially reduced HCD-elevated TG levels. The most efficient results, with a significantly (24.7%; *p* < 0.01) lower TC level, was detected in the Raydel^®^ policosanol-enriched group, followed by (12.3%; *p* < 0.05) reduced TG levels in the BOC Sciences policosanol-enriched group relative to the HCD group.

A notable 2.1-fold (*p* < 0.05) decline in HDL-C was spotted in the HCD-fed group relative to the ND group (Figure 2). The intake of Raydel^®^ policosanol was found to be efficient in enhancing the HDL-C level diminished by the HCD. A considerable 1.8-fold (*p* < 0.05) increase in the level of HDL-C was observed in the Raydel^®^ policosanol-enriched group compared to the HDL-C level in the HCD-fed group. Conversely, the BOC Sciences policosanol did not affect the elevation of HDL-C levels disrupted by the intake of an HCD. Compared to the BOC Sciences policosanol group, a 1.6-fold (*p* < 0.05) elevated HDL-C level in the Raydel^®^ policosanol-enriched group signified the functional superiority of Raydel^®^ policosanol over BOC Sciences policosanol. Consistent with the findings on HDL-C, the HDL-C/TC ratio was significantly (3.6-fold; *p* < 0.001) lower in the HCD-fed group than the ND control group (Figure 2). Consumption of Raydel^®^ policosanol showed significantly higher HDL-C/TC ratios, by 2.6 times (*p* < 0.01) and 2.4 times (*p* < 0.01), respectively, compared to the HCD and BOC Sciences policosanol-supplemented groups. 

In contrast to HDL-C, a substantial 2.2-fold (*p* < 0.001) increase in LDL-C levels was detected in the HCD group compared to the ND group, indicating the influence of an HCD on the enhancement of LDL-C levels (Figure 2). The HCD-induced increase in LDL-C levels was extensively reduced by 1.9 times (*p* < 0.01) by the consumption of Raydel^®^ policosanol. Conversely, the consumption of BOC Sciences policosanol did not impact HCD-elevated LDL-C levels, which were notably 2.1 times (*p* < 0.001) greater than the LDL-C levels detected in the Raydel^®^ policosanol group.

### 2.3. Plasma Hepatic Function Biomarkers

The plasma levels of the hepatic function indicators aspartate aminotransferase (AST) and alanine aminotransferase (ALT) were substantially (~3.0 times; *p* < 0.001) higher in the HCD-fed group than the ND-supplemented group, suggesting the adverse impact of an HCD on hepatic function (Figure 3). Intake of both Raydel^®^ and BOC Sciences policosanol significantly reduced the AST level by 1.9 times (*p* < 0.001) and 1.6 times (*p* < 0.01), and the ALT level by 2.1 times (*p* < 0.001) and 1.7 times (*p* < 0.01), respectively, relative to the HCD-fed group.

### 2.4. Liver Histology

The liver histology (H&E staining) across each group is illustrated in Figure 4A,B. The HCD-fed group displayed substantial hepatic degeneration and high neutrophil accumulation (indicated by the red arrow), which was significantly (7.5 times; *p* < 0.001) in excess of the neutrophil counts detected in the ND group (Figure 4F). The intake of Raydel^®^ and BOC Sciences policosanol substantially affects HCD-impaired liver histology. Significantly lower neutrophil counts, by 4.5 times (*p* < 0.001) and 2.4 times (*p* < 0.01), were observed in the BOC Sciences and Raydel^®^ policosanol groups relative to the HCD-fed group, underscoring the effective role of both policosanols in preventing the hepatic damage caused by the HCD.

ORO staining, as displayed in Figure 4C,G, demonstrated the fatty liver changes in the HCD-fed group, evidenced through the larger ORO-stained area, which is 4.2 times (*p* < 0.001) larger than the basal level as observed in the ND control group. Both Raydel^®^ and BOC Sciences policosanol displayed a significant effect on minimizing the HCD-instigated fatty liver changes, as documented by 3.9-fold (*p* < 0.001) and 1.8-fold (*p* < 0.01) smaller ORO-stained areas than the HCD-fed group. In comparison to BOC Sciences, a 1.9-fold (*p* < 0.01) reduced ORO-stained area was observed in Raydel^®^ policosanol, signifying its higher impact on minimizing the HCD-instigated fatty liver changes.

IHC staining revealed significantly (5.2-fold; *p* < 0.001) elevated production of IL-6 in the HCD-fed group compared to the ND group (Figure 4D,E,H). HCD-induced IL-6 production was significantly (*p* < 0.001) reduced by 4.7 times and 1.9 times (*p* < 0.001) in response to Raydel^®^ and BOC Sciences policosanol consumption. Compared to the BOC Sciences policosanol, a significantly (2.5 times; *p* < 0.001) enhanced efficacy of Raydel^®^ policosanol was observed in curtailing HCD-induced IL-6 production.

### 2.5. Extent of Reactive Oxygen Species (ROS) Production, Apoptosis, and Senescence in the Liver

DHE staining (Figure 5A,D) showed massively (8.2 times; *p* < 0.001) elevated ROS production in the HCD-fed group compared to the ND control group, showing the influence an HCD exerts on ROS production. Consumption of Raydel^®^ and BOC sciences policosanol efficiently curtailed HCD-induced ROS production. Significantly diminished ROS production, by 5.1 times (*p* < 0.001) and 1.5 times (*p* < 0.01), was detected in the Raydel^®^ and BOC Sciences policosanol-supplemented groups compared to the HCD group. The DHE fluorescent intensity in the Raydel^®^ policosanol group was substantially lower by 3.4 times (*p* < 0.001) than the BOC Sciences policosanol, affirming the superior effectiveness of Raydel^®^ policosanol in containing HCD-induced ROS generation.

In alignment with the ROS staining results, the HCD group posed elevated apoptosis levels, as evident by AO fluorescent staining (Figure 5B,E). HCD-induced apoptosis was significantly diminished by 4.3 times (*p* < 0.001) and 1.8 times (*p* < 0.01) through ingestion of Raydel^®^ and BOC Sciences policosanol, respectively. Furthermore, Raydel^®^ policosanol proved to be 2.4 times (*p* < 0.001) more potent than BOC Sciences policosanol in mitigating HCD-induced hepatic apoptosis.

As depicted in Figure 5C,F, the ND control group exhibited a basal level of cellular senescence, marked by the lowest number of SA-β-gal-positive cells, which was significantly (64 times; *p* < 0.001) lower compared to the SA-β-gal-positive cells in the HCD group. Raydel^®^ policosanol consumption significantly mitigated HCD-induced senescence, showing a 6-fold (*p* < 0.001) reduction in the SA-β-gal-stained area relative to the HCD group. Interestingly, BOC Sciences policosanol did not show any effect in reducing HCD-induced cellular senescence in the liver.

### 2.6. Kidney Histology

The ND group (Figure 6A) showed normal kidney histology where well-differentiated proximal (PT) and distal tubules (DT) are arranged intact. In contrast, in the HCD group, a disorganized PT and DT and a basophilic structure (indicated by the blue arrow) corresponding to new nephron generation were observed. Raydel^®^ policosanol substantially mitigated HCD-induced kidney damage, as demonstrated by the densely packed PT and DT regions. Conversely, BOC Sciences policosanol exhibited reduced efficacy in alleviating HCD-induced renal impairment, indicated by the sparsely populated PT and DT regions and the basophilic structure (highlighted by the blue arrow).

DHE staining revealed a notable 4.6-times-augmented (*p* < 0.001) ROS level in the HCD-fed group in contrast to the ND group (Figure 6B,F). The Raydel^®^ policosanol and BOC Sciences policosanol exhibited a marked influence against HCD-elevated ROS levels. Significantly lower levels, by 3.2 times (*p* < 0.001) and 1.2 times (*p* < 0.05), of ROS production were detected in Raydel^®^ and BOC Sciences policosanol-supplemented groups relative to the HCD-fed group. In comparison with BOC Sciences policosanol, the Raydel^®^ policosanol showed higher efficacy as significantly lower ROS production (2.6 times; *p* < 0.001) was spotted in the Raydel^®^ policosanol group than the BOC Sciences policosanol group.

Consistent with the ROS, substantially higher apoptosis was noticed in the HCD-fed group, which was substantially reduced by 3.6 times (*p* < 0.001) and 1.2 times (*p* < 0.05) by the consumption of Raydel^®^ and BOC sciences policosanol (Figure 6C,G). However, the highest anti-apoptotic potential was displayed by Raydel^®^ policosanol, which was markedly (3 times; *p* < 0.001) better than the activity exerted by the BOC Sciences policosanol.

The deposition of lipids in the kidney was examined by ORO staining, which illustrated a significantly (2.2 times; *p* < 0.001) higher lipid accumulation in the HCD-fed group related to the ND control group (Figure 6D,H). Consumption of Raydel^®^ policosanol effectively prevented fat deposition in the kidney, as displayed by a significantly smaller ORO-stained area, by 2.0 times (*p* < 0.001) and 1.5 times (*p* < 0.05), than the HCD and BOC Sciences policosanol-supplemented groups. On the other hand, BOC Sciences policosanol had a non-significant effect (*p* > 0.05) on HCD-provoked fat accumulation in the kidney.

Similar to ORO staining, substantially (2.5 times; *p* < 0.001) higher cellular senescence was detected in the HCD-fed group compared to the ND control group (Figure 6E,I). The consumption of Raydel^®^ policosanol prevented HCD-induced cellular senescence as documented by a significantly (2.4 times; *p* < 0.001) diminished SA-β-gal-stained area compared to the HCD group. The BOC Sciences policosanol displayed a non-significant (*p* > 0.05) impact on HCD-provoked cellular senescence.

### 2.7. Ovary Section Evaluation

Histologic examination (H&E staining) of the ovaries showed a higher abundance of previtellogenic oocytes in the HCD group compared to the group consuming Raydel^®^ policosanol, with a significant 1.2-fold (*p* < 0.001) difference (Figure 7A,E). Contrary to this, no effect of BOC Sciences policosanol was observed on the HCD-instigated previtellogenic oocyte counts. In contrast to the previtellogenic oocytes, a considerably higher (2.9 times; *p* < 0.05) number of early vitellogenic oocytes was detected in the Raydel^®^ policosanol group than in the HCD group, while BOC Sciences policosanol exhibited a non-significant (*p* > 0.05) impact on the early vitellogenic counts impaired by HCD consumption.

DHE staining, as depicted in Figure 7B,F, showed significantly (8.8 times; *p* < 0.001) elevated ROS production in the HCD-fed group relative to the ROS level in the ND control group. The Raydel^®^ and BOC Science policosanol effectively mitigated the ROS generated by the HCD, as manifested by a notably diminished fluorescence intensity of 3.1 times (*p* < 0.001) and 1.6 times (*p* < 0.01) relative to the HCD group (Figure 7B,F). In comparison to BOC Sciences, Raydel^®^ policosanol displayed significantly (1.8 fold; *p* < 0.01) diminished ROS levels, signifying the higher potency of Raydel^®^ policosanol in countering HCD-instigated ROS production.

The HCD-fed group exhibited a significantly higher extent of apoptosis, as evidenced by a 5.2-fold (*p* < 0.001) increase in AO fluorescence intensity compared to the ND control group (Figure 7C,G). Consumption of Raydel^®^ and BOC Sciences policosanol displayed a significant role in preventing HCD-induced apoptosis, evident by significantly diminished apoptosis, by 3.4 times (*p* < 0.001) and 1.4 times (*p* < 0.01), compared to the HCD group. The Raydel^®^ policosanol displayed a remarkable 2.4-fold (*p* < 0.01) reduced AO fluorescence intensity than the BOC Sciences policosanol, attesting to the higher efficacy of Raydel^®^ policosanol with respect to the BOC Sciences policosanol.

In alignment with the results of DHE and AO staining, increased cellular senescence (SA-β-gal-positive cells) was observed in the group that consumed the HCD, which was substantially reduced by 10.6 times (*p* < 0.001) and 1.4 times (*p* < 0.01) with the intake of Raydel^®^ and BOC Sciences policosanol, respectively (Figure 7D,H). Significantly (7.5-fold; *p* < 0.001) lower cellular senescence was noticed in the Raydel^®^ policosanol group than in the BOC Sciences policosanol group. A combined result suggested the significant impact of policosanols, specifically Raydel^®^ policosanol, on protecting ovaries against the adversity posed by the HCD.

### 2.8. Testicular Tissue Evaluation

Histological examination (H&E staining) of the testis exhibited a regular tubular structure with well-differentiated spermatocytes and spermatozoa in the ND group (Figure 8A,E). Compared to this, loosely arranged spermatocytes and spermatozoa with a notable 1.6-fold (*p* < 0.01) enhanced interstitial space between the seminiferous tubules were noticed in the HCD-fed group. The HCD-induced adversity was substantially prevented by consuming Raydel^®^ policosanol, manifested by a significantly (1.7-fold; *p* < 0.01) reduced interstitial space between seminiferous tubules than in the HCD group. In contrast to the Raydel^®^ policosanol, BOC Sciences policosanol did not affect the testis restoration impaired by the consumption of an HCD.

Consumption of both policosanols markedly inhibited HCD-triggered cellular senescence (Figure 8B,F). Significant 5.1-fold (*p* < 0.001) and 2.1-fold (*p* < 0.05) reductions in SA-β-gal-positive cells were noticed in the Raydel^®^ and BOC Sciences policosanol groups compared to the HCD group. In contrast to the BOC Sciences policosanol, a reduced number of SA-β-gal-positive cells (2.4-fold; *p* < 0.01) was noticed in the Raydel^®^ policosanol group, underscoring its higher efficacy over the BOC Sciences policosanol in preventing cellular senescence in the testis.

The DHE and AO fluorescent staining demonstrated a markedly elevated presence of ROS and apoptosis in the HCD group, surpassing that observed in the ND group by a substantial 6.9-fold (*p* < 0.001) and 7.2-fold (*p* < 0.001) margin, respectively (Figure 8C,D,G,H). HCD-induced ROS production was effectively curtailed by the intake of Raydel^®^ and BOC Sciences policosanol, as apparent by significant 3.3-fold (*p* < 0.001) and 2-fold (*p* < 0.001) reductions in DHE fluorescence intensity corresponding to ROS production in Raydel^®^ and BOC Sciences policosanol groups relative to the HCD group. Similarly, the AO staining showed remarkable reductions of 3.2-fold (*p* < 0.001) and 1.4-fold (*p* < 0.001) in apoptosis in the Raydel^®^ and BOC Sciences policosanol groups relative to the HCD group. When compared to BOC Sciences, Raydel^®^ policosanol showed significantly (1.7-fold (*p* < 0.001) and 1.4-fold (*p* < 0.001)) higher efficacy in mitigating HCD-provoked ROS production and apoptosis, respectively, attesting to the functional superiority of Raydel^®^ policosanol.

### 2.9. Egg-Laying Ability in Response to Policosanol Consumption

The ingestion of HCD had a noteworthy impact on the egg-laying activity of zebrafish (Figure 9A). The HCD-fed zebrafish produced significantly (3.5-fold (*p* < 0.05)) fewer eggs than the ND control group. Intake of Raydel^®^ and BOC Sciences policosanol substantially elevated the HCD-impaired egg-laying behavior of zebrafish. A notable 7.3-fold (*p* < 0.01) and 3.3-fold (*p* < 0.05) increase in egg production was perceived in the Raydel^®^ and BOC Sciences policosanol groups with respect to the HCD group. Surprisingly, the Raydel^®^ policosanol showed significantly (~2-fold (*p* < 0.05)) enhanced egg-laying efficacy compared to the ND and BOC sciences policosanol groups.

Further, the survivability of the laid eggs across all the groups was monitored up to 120 h post-fertilization (hpf). Figure 9B illustrates a consistent decrease in embryo survival in the HCD group, starting at 24 hpf (79.7%) and declining to 49.1% at 120 hpf. A similar trend was observed for the embryos in the BOC Sciences policosanol-supplemented group, displaying 52.2% embryo survivability at 120 hpf. Contrary to this, the embryos from the Raydel^®^ policosanol group showed a substantial ~1.4-fold (*p* < 0.01) elevated survivability at 120 hpf compared to the BOC Sciences and HCD-fed groups.

A substantial teratogenic manifestation was detected in the HCD group, where most of the embryos (~60%) appeared with a minor developmental defect with respect to tail fin curvature, yolk sac edema, pericardial edema, and least somite counts (29 ± 1) (Figure 9C). Likewise, most embryos (~43%) in the BOC Sciences policosanol group displayed developmental deformities, mainly back bending with average somite counts of 31 ± 3. Contrary to this, the majority of the viable embryos (~87%) in the Raydel^®^ policosanol group demonstrated no severe developmental deformities and appeared with average somite counts of 35 ± 2, analogous to the ND control group. The collective outcomes demonstrate the marked influence of Raydel^®^ policosanol in countering HCD-induced disruption in egg-laying and embryonic development, underscoring its positive effect on the reproductive health of zebrafish.

## 3. Discussion

In this study, the comparative efficacy of two distinct policosanols derived from similar source materials (i.e., sugarcane wax) but originating from different locations was tested in vivo by employing a zebrafish model. This study is the continuation of our earlier findings, where the Raydel^®^ and BOC Sciences policosanol entrapped in reconstituted high-density lipoprotein (rHDL) deciphered substantial variations in their antioxidant, antiglycation, wound-healing, and anti-inflammatory activity [35]. To broaden these earlier insights in the present study, the comparative effect of Raydel^®^ and BOC Sciences policosanol consumption on hyperlipidemic adult zebrafish was tested in terms of survivability, body weight, plasma lipid profile, and the functionality of the organs. The selection of zebrafish as a model organism in the present study is based on the high genetic similarity of zebrafish and humans [36], which renders it a suitable model for the preclinical [37] and toxicological [38] analysis of drugs and nutraceuticals. Several reports suggest that the outcomes obtained from preclinical studies in zebrafish successfully laid the foundation for some small molecules to promote human clinical trials [37], underscoring zebrafish’s relevance as an important organism for preclinical studies. Additionally, lipid metabolism in zebrafish mimics that of humans owing to the presence of critical receptors, different lipid metabolism enzymes, lipoproteins, and apolipoproteins [39]. More importantly, like humans, zebrafish possess cholesteryl ester transfer protein (CETP) activity, which plays a key role in lipid metabolism to lower HDL-C [39]. In contrast to this, mice and rats do not have CETP activity and thus zebrafish are a canonical model for lipoprotein-related research [39].

Zebrafish subjected to Raydel^®^ and BOC Sciences policosanol for 12 weeks demonstrated a survival rate akin to the ND control group, underscoring the safety of both policosanols. Slightly reduced survivability was observed in the ND control group compared to the HCD-only group and the groups supplemented with Raydel^®^ or BOC Sciences policosanol; however, these findings are not unusual as the difference between the groups is non-significant. The Raydel^®^ and BOC Sciences policosanol displayed a minor effect in alleviating the HCD-induced body weight enhancement. These results slightly differed from the outcomes of previous studies documenting the substantial impact of Raydel^®^ policosanol on body weight management altered by the consumption of HCD [3]. The basic reason for the disparity in results is the low amount (0.1% *w*/*w*) of policosanol consumed in the present study, which is ten times lower than the previously consumed amount (1% *w*/*w*) that effectively mitigated the body weight gain caused by HCD [3].

Policosanol is a nutraceutical that is known to maintain the blood lipid profile by effectively alleviating TC and TG levels [1,10,40]; however, the activities of different policosanols vary significantly [3,34]. Consistently, we also noticed the impactful role of Raydel^®^ and BOC Sciences policosanol in attenuating HCD-induced TC and TG levels. However, compared to the BOC Sciences policosanol, Raydel^®^ policosanol displayed a significantly more pronounced effect in reducing the HCD-elevated TC and TG levels. The variation in the activity between BOC Sciences and Raydel^®^ policosanol is due to the difference in their LCAA composition, which aligns well with the earlier studies deciphering that policosanol’s blood lipid-lowering effect depends on LCAA composition and source material [6,34]. In a recent study, five distinct commercial policosanol brands showed a substantial variation in blood lipid profile primarily due to the composition of the LCAAs in these policosanol brands [3].

Octacosanol, a primary LCAA in policosanol responsible for a variety of biological activities [41] including a lipid-lowering effect [42], was found to be slightly higher in the BOC Sciences policosanol than the Raydel^®^ policosanol [35]. Irrespective of this, the Raydel^®^ policosanol displayed a significantly greater effect than the BOC Sciences policosanol in reducing the HCD-induced TC and TG levels. The better activity of the Raydel^®^ policosanol compared to the BOC Sciences policosanol may be strongly influenced by a 7.5-fold and 3-fold higher prevalence of triacontanol and hexacosanol in it (compared to the BOC Sciences policosanol) [35], which either work alone or synergistically with the other LCAAs to alleviate the HCD-induced TC and TG levels profoundly. The notion is convincingly endorsed by the studies documenting the inhibitory role of triacontanol [43] and hexacosanol [44] on cholesterol synthesis. The study documented the significant role of hexacosanol in the reduction of TC and TG by allosterically regulating the activation of AMP-activated protein kinase (AMPK), which subsequently inhibits the activity of 3-hydroxy-3-methyl-glutaryl-coenzyme A reductase (HMG-CoA), a principal rate-limiting enzyme for cholesterol biosynthesis [44]. Moreover, hexacosanol has a ~1.6-fold lower effective concentration (EC_50_) than octacosanol towards AMPK, suggesting the functional superiority of hexacosanol in activating AMPK and consequently restraining cholesterol biosynthesis via inhibiting HMG-CoA activity [44]. These findings strengthen the current outcomes, showing that the higher prevalence of hexacosanol in the Raydel^®^ policosanol is a key contributor towards its functional superiority over the BOC Sciences policosanol to alleviate TC and TG levels. Apart from the AMPK-mediated inhibition of HMG-CoA, hexacosanol also impacts the transcriptional regulation of HMG-CoA by inhibiting the nuclear transition of sterol regulatory element-binding protein-2 (SREBP-2), an important regulatory factor for the transcription of HMG-CoA [44], hence affecting cholesterol biosynthesis effectively.

Similar to the TC and TG levels, the Raydel^®^ policosanol curtailed the HCD-induced LDL-C; however, the BOC Sciences policosanol was found to be ineffective in reducing LDL-C. The difference in activity lies in the compositional difference between the two policosanols, which is consistent with the previous findings implying that the differences in policosanol activity are based on their composition [6,34]. The ability of Raydel^®^ policosanol to reduce LDL-C is due to its unique composition of LCAAs, especially the higher prevalence of hexacosanol, which has been found to be effective in reducing LDL-C levels [44]. Several clinical and preclinical studies have documented the impact of policosanol consumption on the improvement of HDL-C levels [10,45,46,47]. Consistent with these findings, Raydel^®^ policosanol substantially elevated HDL-C levels. Contrary to this, the BOC Sciences policosanol was recognized as ineffective in elevating HDL-C levels. This variation in activity is associated with differences in the composition of LCAAs between the policosanols, which has been supported by a preceding study where similar results against carboxymethyllysine (CML)-impaired HDL-C levels were observed by rHDL containing BOC Sciences and Raydel^®^ policosanol [35], signifying the effectiveness of Raydel^®^ policosanol over BOC Sciences policosanol.

HCD provokes an adverse effect on the liver [48], leading to hepatic degeneration, neutrophil infiltration, fatty liver changes, and inflammation. Hepatic histology suggests the effective role of both Raydel^®^ and BOC Sciences policosanol in preventing hepatic impairment caused by the consumption of an HCD. Consistent with hepatic histology, the serum hepatic function biomarkers AST and ALT also suggest a hepatoprotective role of policosanol against HCD-induced damage. However, compared to the BOC Sciences policosanol, Raydel^®^ policosanol showed significantly higher efficacy in preventing HCD-provoked fatty liver changes and IL-6 generation. A compositional difference in LCAAs between the two policosanols is recognized as a key contributor to this disparity. Specifically, the substantially higher prevalence of hexacosanol in the Raydel^®^ policosanol possibly has a pivotal effect against fatty liver changes, which is in accordance with earlier reports demonstrating a decisive inhibitory effect of hexacosanol on hepatic lipid accumulation [44]. The induction of autophagy has been recognized as an important event in inhibiting hepatic lipid accumulation and preventing steatosis (fatty liver changes) [49]. A visible effect of hexacosanol on the autophagy induction mediated by the upregulation of autophagy-related genes (ATG16L) and anti-microtubule-associated protein 1A/1B light chain 3 (LC3-II) has been documented to aid in mitigating hepatic lipid accumulation and, consequently, fatty liver changes [44], supporting the present outcomes concerning the inhibition of fatty liver changes in response to Raydel^®^ policosanol. HCD-induced ROS generation and apoptosis are substantially prevented by the consumption of both BOC Sciences and Raydel^®^ policosanol. However, Raydel^®^ policosanol was found to be more effective than BOC Sciences in inhibiting HCD-provoked production of ROS generation and apoptosis in the liver. The distinct composition of Raydel^®^ and BOC Sciences policosanol is the major driving factor for the difference in activity. A substantial 7.5-fold higher amount of triacontanol in Raydel^®^ policosanol compared to the BOC Sciences policosanol [35] may lead to superior activity in containing ROS production as the effective role of triacontanol is documented to prevent oxidative stress [1]. In a recent investigation, Raydel^®^ policosanol encapsulated in rHDL demonstrated significantly higher antioxidant activity to prevent LDL oxidation and CML-induced ROS generation in embryos and adult zebrafish than the BOC Sciences policosanol, strengthening the current findings and establishing the higher efficacy of Raydel^®^ policosanol in containing HCD-triggered ROS production [35]. The AO staining revealed the significantly higher efficacy of Raydel^®^ policosanol than the BOC Sciences policosanol in preventing HCD-induced apoptosis. The better activity of Raydel^®^ over BOC Sciences policosanol is due to the higher impact of Raydel^®^ policosanol in inhibiting cellular ROS production, which has been accredited as a major culprit for the induction of apoptosis [50,51]. These results aligned with the earlier findings that documented the correlation between ROS production and apoptosis in distinct policosanol [3]. No effect of BOC Sciences policosanol was observed on cellular senescence; however, Raydel^®^ policosanol significantly altered HCD-induced cellular senescence. We speculate that the low ROS production in response to Raydel^®^ policosanol is the major event in inhibiting cellular senescence, as ROS-induced oxidative stress has been documented as the major culprit causing cellular senescence [52,53].

Dyslipidemia caused by the consumption of an HCD has been observed to impair the functionality of the kidneys and reproductive system [54,55,56]. The accumulating literature has demonstrated the impact of policosanol on preventing kidney and other vital organ damage [57]. However, the studies suggest a great disparity in protective activity based on the composition of LCAAs in the policosanol [6,34]. The presence of hexacosanol was found to be competent in averting kidney impairment by modulating the expression of nitric oxide synthase [58] and protein kinase C (PKC) [59], leading to the inhibition of glomerular sclerosis and improvement of diabetic neuropathy [58,59]. Herein, a disparity was observed between the Raydel^®^ and BOC Sciences policosanols in preventing kidney damage by averting HCD-induced ROS generation, apoptosis, lipid accumulation, and cellular senescence. As compared to the BOC Sciences policosanol, the Raydel^®^ policosanol displayed significantly superior activity in preventing HCD-induced kidney damage owing to its unique combination of LCAAs, specifically a 3.5-fold higher prevalence of hexacosanol [35], whose nephron protective role is well-described [58].

In addition, the Raydel^®^ policosanol displayed a significantly better effect over the BOC Sciences policosanol on the restoration of HCD-provoked testis, ovary impairment, ROS generation, and cellular senescence. In addition, a higher egg-laying ability, as well as the survivability of eggs with minimum teratogenic defects, was observed in the Raydel^®^ policosanol-fed zebrafish, signifying its positive impact on reproductive health. The results are consistent with our earlier findings, where we observed the positive impact of Cuban policosanol (Raydel^®^) on the egg-laying behavior of zebrafish, which was significantly better than the impact exerted by the policosanol from China and the USA [6]. However, there is limited research addressing the role of policosanol in reproductive health [6,60] and this needs to be explored in detail. However, some preliminary studies have deciphered the potential of octacosanol on reproductive health via modulation of progesterone, luteinizing and follicle-stimulating hormone synthesis, and egg production [61]. The presence of octacosanol in both policosanols seems to be behind their impact on reproductive health. The better activity of the Raydel^®^ policosanol over the BOC Sciences policosanol can be justified by the higher prevalence (≥3.5 fold) of tetratriacontanol, dotriacontanol, triacontanol, and hexacosanol, which synergistically act with octacosanol; however, the exact reason behind this needs to be explored.

## 4. Materials and Methods

### 4.1. Materials

Raw materials of Raydel^®^ policosanol were provided by the National Center for Scientific Research (CNIC, Havana, Cuba). Best of Chemical (BOC) Sciences policosanols were procured from BOC Sciences (Shirley, NY, USA). The Raydel^®^ and BOC Sciences policosanols were extracted and purified from sugarcane wax from Cuba and China, respectively [35]. A detailed specification of the long-chain aliphatic alcohol (LCAA) proportions in the Raydel^®^ and BOC science policosanols [35] is listed in Table 1. All remaining chemicals and reagents utilized in this study were of analytical grade and used as received. For detailed specifications, refer to Appendix A.

### 4.2. Formulation of High-Cholesterol Diet (HCD) Enriched with Policosanol

The normal tetrabit (ND, Tetrabit GmbH D49304, Melle, Germany), a regular zebrafish diet, was blended with cholesterol (4% *w*/*w*, final) to prepare the high-cholesterol diet (HCD). The Raydel^®^ policosanol and BOC Sciences policosanol (0.1% *w*/*w*, final) were mixed individually with HCD to prepare two distinct policosanols supplemented with HCD diets.

### 4.3. Zebrafish Culture and Feeding with High-Cholesterol Diet (HCD)

Adult zebrafish (19 weeks old, *n* = 224) were maintained at 28 °C with constant aeration of water under 14 h/10 h light and dark photoperiods, respectively, following the standard guidelines for Animal Care and Use as prescribed by the Raydel^®^ Research Institute (approval code RRI-20-004, approval date 3 January 2020). The zebrafish were maintained separately on two distinct diets, i.e., ND (*n* = 56) and HCD (*n* = 168), for nine weeks under the abovementioned culture conditions.

### 4.4. Zebrafish Feeding with High-Cholesterol Diet (HCD) Enriched with Policosanol

After feeding for nine weeks, the ND (*n* = 56)-fed zebrafish were further maintained at ND for the next 12 weeks (and served as a control group). At the same time, the HCD (*n* = 168)-fed zebrafish were segregated into three distinct groups. The zebrafish (*n* = 56) in group I were maintained on the HCD, while zebrafish in group II (*n* = 56) and group III (*n* = 56) were fed with HCD supplemented with Raydel^®^ policosanol and BOC Sciences policosanol, respectively, for the next 12 weeks.

During 12 weeks of supplementation, mortality and body weight were periodically monitored across all the groups. Body weight in the distinct groups was quantified gravimetrically after anesthetizing zebrafish by submerging them in 2-phenoxyethanol solution (0.1% *v*/*v*) for 60 sec, following immediate body weight analysis using an electronic weighing balance (Ohaus, Parsippany-Troy Hills, NJ, USA).

### 4.5. Blood Analysis

After 12 weeks of supplementation, all zebrafish groups underwent euthanasia via hypothermic shock [62], followed by immediate blood collection for plasma extraction. The plasma samples were analyzed for total cholesterol (TC) and triglycerides (TG), high-density-lipoprotein cholesterol (HDL-C), aspartate transaminase (AST), and alanine transaminase (ALT) quantification, utilizing commercial kits as per the manufacturer’s guidelines. Detailed procedures for the plasma analysis are described in Appendix A. Low-density-lipoprotein cholesterol (LDL-C) levels were determined using the Friedwald equation [TC − HDL-C − (TG/5)].

### 4.6. Histological Analysis

Distinct organs (liver, kidney, ovary, and testis) from the sacrificed zebrafish were removed surgically and stored in a 10% formalin solution. The tissue was preserved in the refrigerator for further use. For the histological analysis, different tissue sections were dehydrated with ethanol and fixed in paraffin wax, followed by 7 μm thick sectioning using a microtome. The tissue section (liver, kidney, ovary, and testis) was individually processed for Hematoxylin and Eosin (H&E) staining, following the earlier described protocol [63].

Oil red O staining (ORO) was conducted according to the method previously outlined [62]. Briefly, 7 μm thick tissue sections were treated with an ORO solution and incubated at 60 °C. After 5 min, the excess stain was removed with 60% ethanol, followed by counterstaining with Hematoxylin for 30 s. The stained section was washed with water and observed under the microscope.

### 4.7. Detection of Interleukin (IL-6) by Immunohistochemistry (IHC)

IHC was performed to detect IL-6 levels in the hepatic tissue following the earlier described method [64]. The hepatic tissue section (7 μm thick) was covered with 200-fold diluted IL-6-specific primary antibodies (ab9324, Abcam, London, UK) at a low temperature (4 °C). After overnight incubation, the unbounded antibodies were rinsed, and the section was developed using an EnVison + system-HRP polymer kit (Dako, Glostrup, Denmark) containing a chromogenic substrate along with an anti-IL-6 antibody-specific HRP-conjugated secondary antibody.

### 4.8. Fluorescent Staining for ROS Production and Apoptosis

Fluorescent staining of dihydroethidium (DHE) and acridine orange (AO) in the tissue section was conducted according to a previously established protocol [62]. Briefly, the tissue section was treated with 250 μL of DHE (30 μM) and incubated for 30 min at room temperature (RT) in darkness. After three washes with phosphate-buffered saline (PBS), the stained sections were visualized under a fluorescent microscope using excitation/emission wavelengths of 585 nm and 615 nm, respectively. For AO staining, 250 μL of AO solution (5 μg/mL) was applied to the tissue sections and incubated for 30 min at RT in darkness. Following three PBS washes, the AO-stained sections were examined under a fluorescent microscope using excitation/emission wavelengths of 505 nm and 535 nm.

### 4.9. Senescent-Associated β-Galactose Staining

Cellular senescence in the tissue section was assessed using senescent-associated β-galactosidase staining, following the previously described method [65]. Briefly, tissue sections were fixed with 4% paraformaldehyde (5 min), washed with PBS, and then treated with X-gal solution (0.1% in citrate buffer, pH 5.9) for 16 h at RT. After washing with PBS, the section was examined microscopically to visualize senescent cells.

### 4.10. Statistical Analysis

To determine statistical significance between the groups, one-way ANOVA followed by Tukey’s post hoc testing was performed using the SPSS software (version 23.0, Inc., Chicago, IL, USA). Mean values ± standard error of the mean (SEM) from three distinct experiments are depicted in the graphs.

## 5. Conclusions

The functional disparity of in vivo consumption between Raydel^®^ and BOC Sciences policosanols has been established in the current study. The Raydel^®^ policosanol proved highly effective in curtailing TC, TG, and LDL-C along with a substantial elevation of HDL-C levels impaired by the HCD. In addition, a superior effect of the Raydel^®^ policosanol was noticed in preventing HCD-induced liver, kidney, and testis damage. Furthermore, the Raydel^®^ policosanol exhibited a noteworthy influence on the reproductive health of zebrafish by modulating HCD-induced egg-laying activity and teratogenic effects. This study affirms the higher efficacy of Raydel^®^ policosanol over the BOC sciences policosanol owing to its unique combination of LCAAs, which serves as a reference for formulating the policosanol composition to achieve optimum beneficial effects.

## Figures and Tables

**Figure 1 pharmaceuticals-17-01103-f001:**
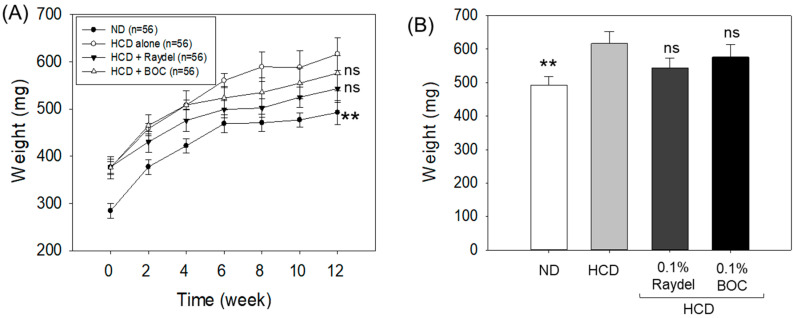
Body weight of zebrafish consuming a high-cholesterol diet (HCD) supplemented with Raydel^®^ and BOC Sciences policosanol. (**A**) Zebrafish body weight at different time points (0–12 weeks). (**B**) Zebrafish body weight at 12 weeks. Values depicted in the graphs are the mean ± SEM of three assorted studies. ND, normal diet; HCD, high-cholesterol diet (i.e., ND containing 4% cholesterol), while HCD + Raydel or BOC represent the HCD + 0.1% Raydel^®^ policosanol or BOC sciences policosanol, respectively. Significance level of *p* < 0.01 is indicated by ** with respect to the HCD group. ns, non-significant.

**Figure 2 pharmaceuticals-17-01103-f002:**
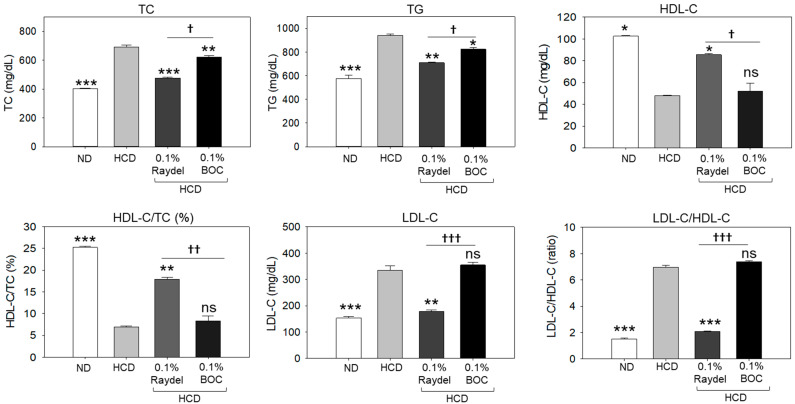
Plasma lipid profile of zebrafish consuming a high-cholesterol diet (HCD) supplemented with Raydel^®^ and BOC Sciences policosanol for 12 weeks. Values depicted in the graphs are the mean ± SEM of three assorted studies. ND, normal diet; HCD, high-cholesterol diet (i.e., ND containing 4% cholesterol), while HCD + Raydel or BOC represent the HCD + 0.1% Raydel^®^ policosanol or BOC Sciences policosanol, respectively. Significance levels of *p* < 0.05, *p* < 0.01, and *p* < 0.001 are denoted by *, **, and *** with respect to the HCD group. ^†^, ^††^, and ^†††^ depict significance levels at *p* < 0.05, *p* < 0.01, and *p* < 0.001, respectively, between Raydel^®^ and BOC Sciences policosanol groups. ns, non-significant.

**Figure 3 pharmaceuticals-17-01103-f003:**
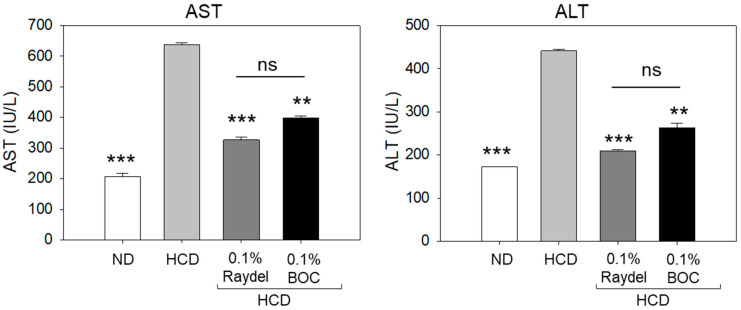
Levels of the hepatic function biomarkers aspartate aminotransferase (AST) and alanine aminotransferase (ALT) in zebrafish consuming a high-cholesterol diet (HCD) supplemented with Raydel^®^ and BOC Sciences policosanol for 12 weeks. Values depicted in the graphs are the mean ± SEM of three assorted studies. ND, normal diet; HCD, high-cholesterol diet (i.e., ND containing 4% cholesterol), while HCD + Raydel or BOC represent the HCD + 0.1% Raydel^®^ policosanol or BOC Sciences policosanol, respectively. Significance levels of *p* < 0.01 and *p* < 0.001 are denoted by ** and *** with respect to the HCD group. ns, non-significant.

**Figure 4 pharmaceuticals-17-01103-f004:**
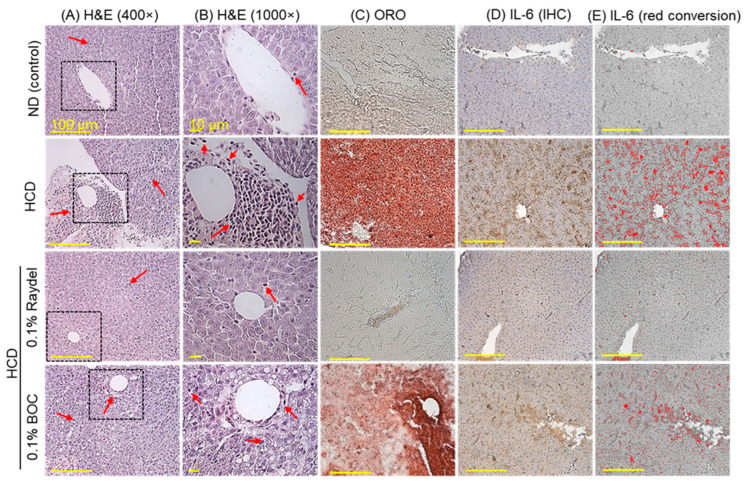
Hepatic histology of zebrafish consuming a high-cholesterol diet (HCD) supplemented with Raydel^®^ and BOC Sciences policosanol for 12 weeks. (**A**,**B**) Hematoxylin and eosin (H&E) staining at 400× and 1000× magnification, respectively. Red arrow indicates infiltrated neutrophil. (**C**) Oil red O (ORO) staining. (**D**) Immunohistochemistry (IHC) for the detection of IL-6. (**E**) The brown color IL-6-stained area is interchanged with red color (to enhance visibility) at the brown color threshold values of 20–120 utilizing Image J software, version 1.53r). (**F**) Percentage of neutrophils quantified in the H&E section. (**G**) Image J-based assessment of ORO-stained areas. (**H**) Image J-based IL-6-stained areas. Values depicted in the graphs are the mean ± SEM of three assorted studies. ND, normal diet; HCD, high-cholesterol diet (i.e., ND containing 4% cholesterol), while HCD + Raydel or BOC represent the HCD + 0.1% Raydel^®^ policosanol or BOC Sciences policosanol, respectively. Significance levels of *p* < 0.01 and *p* < 0.001 are denoted by ** and *** with respect to the HCD group, while ^††^ and ^†††^ depict significance levels at *p* < 0.01 and *p* < 0.001, respectively, between Raydel^®^ and BOC Sciences policosanol groups. “ns” denotes non-significant differences between the groups.

**Figure 5 pharmaceuticals-17-01103-f005:**
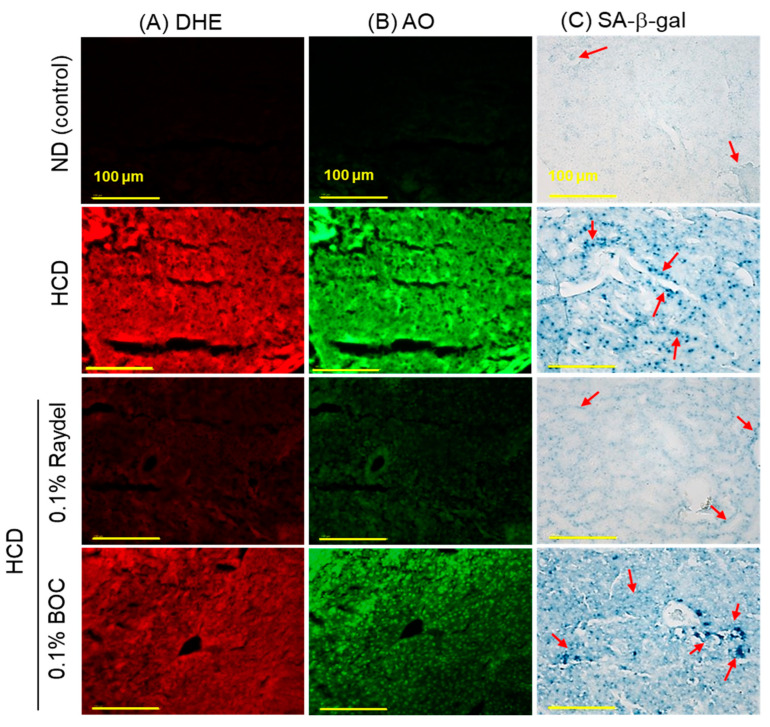
Dihydroethidium (DHE), acridine orange (AO), and senescence-associated β-galactosidase (SA-β-gal) staining in the liver of zebrafish following 12 weeks of consumption of a high-cholesterol diet (HCD) supplemented with Raydel^®^ and BOC Sciences policosanol. (**A**) DHE fluorescent staining for the detection of reactive oxygen species (ROS). (**B**) AO fluorescent staining to show the magnitude of apoptosis. (**C**) SA-β-gal staining to ascertain cellular senescence; the red arrow indicates the SA-β-gal-positive cells [100 μm, scale bar]. (**D**) and DHE fluorescence intensity quantification by Image J software, version 1.53r. (**E**) AO fluorescence intensity quantification by the Image J software. (**F**) Percentage quantification of SA-β-gal-stained area. Values depicted in the graphs are the mean ± SEM of three assorted studies. ND, normal diet; HCD, high-cholesterol diet (i.e., ND containing 4% cholesterol), while HCD + Raydel or BOC represent the HCD + 0.1% Raydel^®^ policosanol or BOC Sciences policosanol, respectively. Significance levels of *p* < 0.01 and *p* < 0.001 are denoted by ** and *** with respect to the HCD group, while ^†††^ depicts significance levels at *p* < 0.001 between Raydel^®^ and BOC Sciences policosanol groups. “ns” denotes non-significant differences between the groups.

**Figure 6 pharmaceuticals-17-01103-f006:**
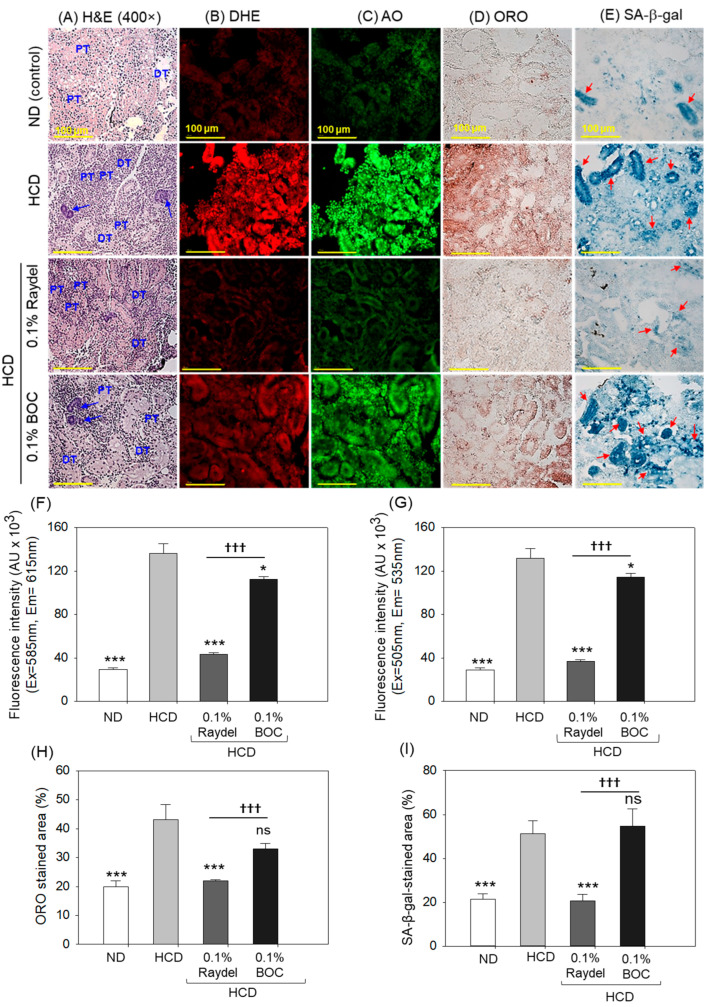
Kidney histology of zebrafish consuming a high-cholesterol diet (HCD) supplemented with Raydel^®^ and BOC Sciences policosanol for 12 weeks. (**A**) The distal tubule (DT) and proximal tubule (PT) are labeled, with the blue arrow highlighting the basophilic cluster (dark-stained) representing the new nephrons. (**B**,**C**) Dihydroethidium (DHE) and acridine orange (AO) fluorescent staining for the detection of reactive oxygen species (ROS) and apoptosis, respectively. (**D**) Oil red O (ORO) staining. (**E**) Senescent-associated β-galactosidase (SA-β-gal) staining to determine cellular senescence; the red arrow implies SA-β-gal-positive cells. [100 μm, scale bar]. (**F**) DHE fluorescence intensity quantification by Image J software. (**G**) AO fluorescence intensity quantification by Image J software. (**H**) Image J software-based assessment of ORO-stained areas. (**I**) Image J software-based assessment of SA-β-gal-stained areas. ND, normal diet; HCD, high-cholesterol diet (i.e., ND containing 4% cholesterol), while HCD + Raydel or BOC represent the HCD + 0.1% Raydel^®^ policosanol or BOC Sciences policosanol, respectively. Significance levels of *p* < 0.05 and *p* < 0.001 are denoted by * and *** with respect to the HCD group, while ^†††^ depicts significance levels at *p* < 0.001 between Raydel^®^ and BOC Sciences policosanol groups. “ns” denotes non-significant differences between the groups.

**Figure 7 pharmaceuticals-17-01103-f007:**
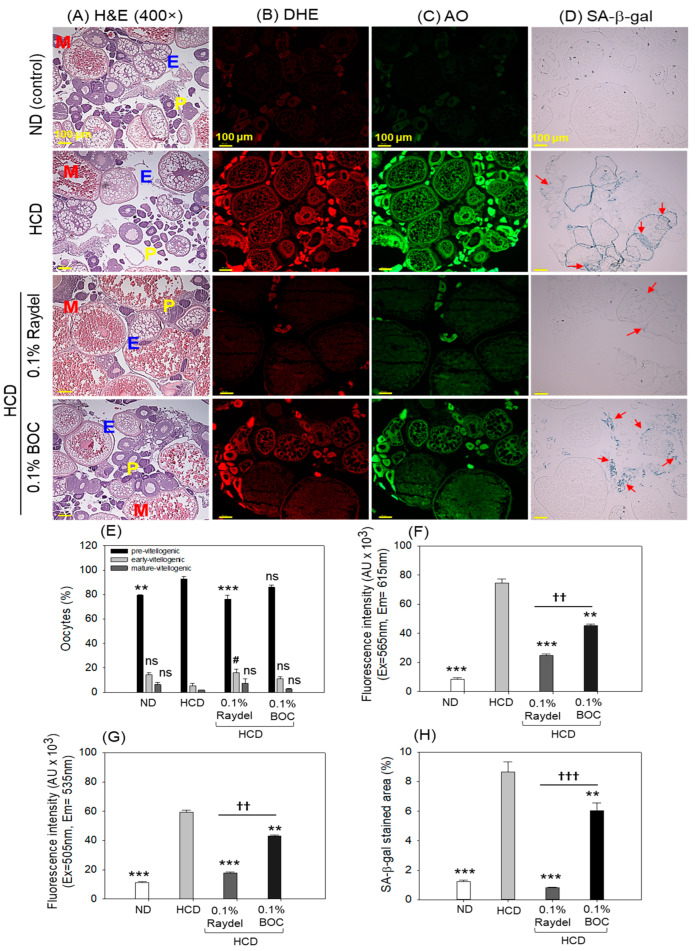
Ovary histology of zebrafish consuming a high-cholesterol diet (HCD) supplemented with Raydel^®^ and BOC Sciences policosanol for 12 weeks. (**A**) Hematoxylin and Eosin (H&E) staining; P, E, and M represent pre, early, and mature vitellogenic oocytes, respectively. (**B**,**C**) Dihydroethidium (DHE) and acridine orange (AO) fluorescent staining for the identification of reactive oxygen species (ROS) and apoptosis, respectively. (**D**) Senescent-associated β-galactosidase (SA-β-gal) staining to determine cellular senescence; the red arrow highlights the SA-β-gal-positive cells. (**E**) Oocyte count in the ovary. (**F**) Image J-based quantification of DHE fluorescent intensities. (**G**) Image J-based quantification of AO fluorescent intensities. (**H**) Percentage quantification of SA-β-gal-stained area. ND, normal diet; HCD, high-cholesterol diet (i.e., ND containing 4% cholesterol), while HCD + Raydel or BOC represent the HCD + 0.1% Raydel^®^ policosanol or BOC Sciences policosanol, respectively. Significance levels of *p* < 0.01 and *p* < 0.001 are denoted by ** and *** with respect to the HCD group, while ^††^ and ^†††^ depict significance levels at *p* < 0.01 and *p* < 0.001, respectively, between Raydel^®^ and BOC Sciences policosanol groups. The # represents the significance level at *p* < 0.05 denoted by # for the early vitellogenic oocytes counts with respect to the HCD group. The “ns” denotes non-significant differences between the groups.

**Figure 8 pharmaceuticals-17-01103-f008:**
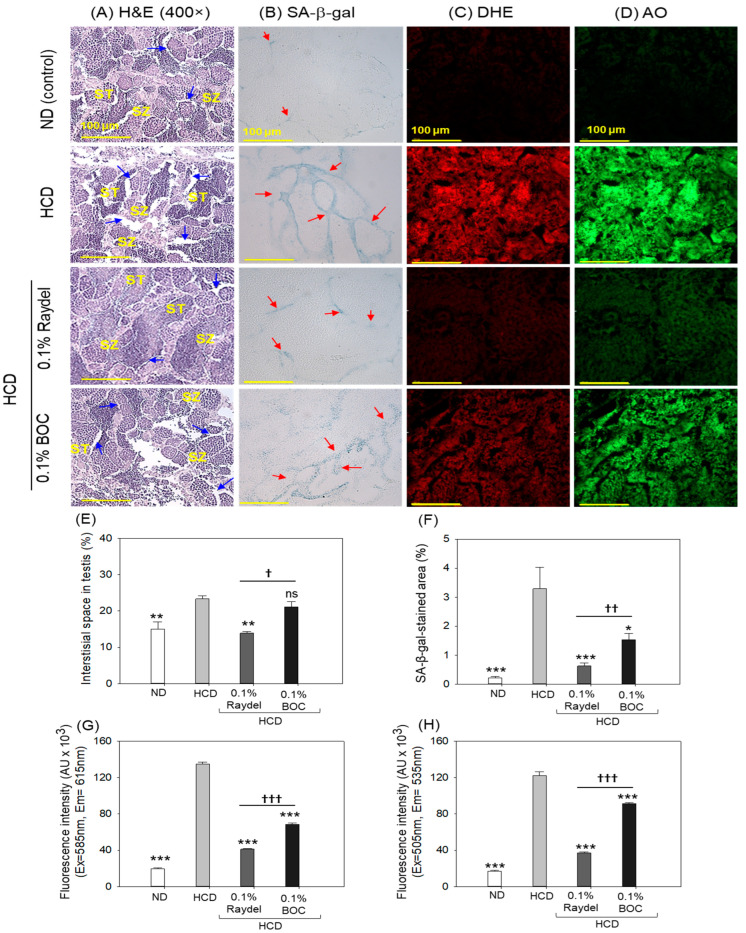
Testis histology of zebrafish consuming a high-cholesterol diet (HCD) supplemented with Raydel^®^ and BOC Sciences policosanol for 12 weeks. (**A**) Hematoxylin and Eosin (H&E) staining; spermatocytes and spermatozoa are abbreviated by ST and SZ; the blue arrow depicts the interstitial space within the seminiferous tubules. (**B**) Senescent-associated β-galactosidase (SA-β-gal) staining to determine cellular senescence; the red arrow indicates the SA-β-gal-positive cells. (**C**,**D**) Dihydroethidium (DHE) and acridine orange (AO) fluorescent staining for the detection of reactive oxygen species (ROS) and apoptosis, respectively. (**E**) Image J-based assessment of interstitial space and SA-β-gal-stained area. (**F**) Percentage quantification of SA-β-gal-stained area. (**G**,**H**) DHE and AO fluorescence intensity, respectively quantified by Image J software. ND, normal diet; HCD, high-cholesterol diet (i.e., ND containing 4% cholesterol), while HCD + Raydel or BOC represent the HCD + 0.1% Raydel^®^ policosanol or BOC Sciences policosanol, respectively. Significance levels of *p* < 0.05, *p* < 0.01, and *p* < 0.001 are denoted by *, **, and *** with respect to the HCD group, while ^†^, ^††^, and ^†††^ depict significance levels at *p* < 0.05, *p* < 0.01, and *p* < 0.001, respectively, between Raydel^®^ and BOC Sciences policosanol groups. The “ns” denotes non-significant differences between the groups.

**Figure 9 pharmaceuticals-17-01103-f009:**
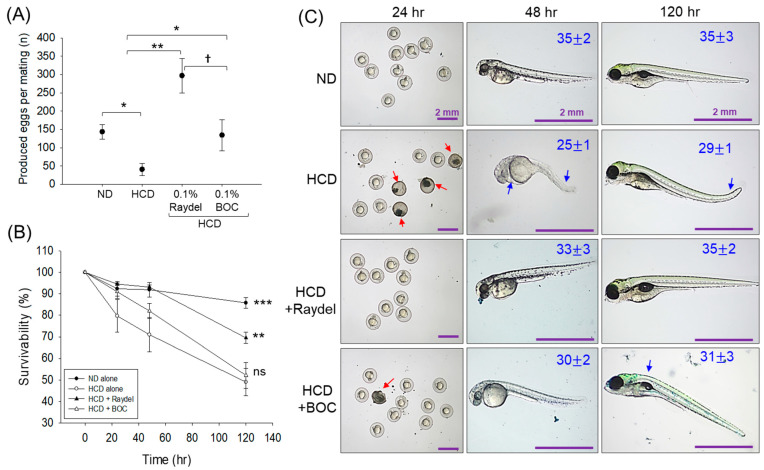
Egg-laying ability of zebrafish following 12 weeks of consumption of a high-cholesterol diet (HCD) supplemented with Raydel^®^ and BOC Sciences policosanol. (**A**) Average number of produced eggs. (**B**) Survivability of the eggs during 120 hpf. (**C**) Morphology of the developed embryos at 24 h, 48 h, and 120 h post-fertilization; average somite counts are denoted as numeric values in blue; red and blue arrows denote embryo death and structural deformities. ND, normal diet; HCD, high-cholesterol diet (i.e., ND containing 4% cholesterol), while HCD + Raydel^®^ or BOC represent the HCD + 0.1% Raydel^®^ policosanol or BOC Sciences policosanol, respectively. Significance levels of *p* < 0.05, *p* < 0.01, and *p* < 0.001 are denoted by *, **, and *** with respect to the HCD group. ^†^ (*p* < 0.05) depicts the significance level between the Raydel^®^ and BOC Sciences policosanol groups. ns, non-significant.

**Table 1 pharmaceuticals-17-01103-t001:** Description and composition of policosanols from Raydel and BOC Sciences [35].

Description/Composition of Long-Chain Aliphatic Alcohols	Raydel Policosanol	BOC Policosanol
Source material and country of origin	Sugarcane wax from Cuba	Sugarcane wax from China
Manufacturer	CNIC, Cuba	BOC Sciences, USA
Average molecular weight	418	411
Composition of long-chain aliphatic alcohols (LCAAs) (mg/g) (%)		
1-tetracosanol (C_24_H_50_O)	0.3 (0.0)	4.0 (0.5)
1-hexacosanol (C_26_H_54_O)	38.0 (3.9)	11.0 (1.2)
1-heptacosanol (C_27_H_56_O)	9.0 (0.9)	21.0 (2.3)
1-octacosanol (C_28_H_58_O)	692.0 (70.5)	819.0 (90.5)
1-nonacosanol (C_29_H_60_O)	6.0 (0.6)	12.0 (1.4)
1-triacontanol (C_30_H_62_O)	139.0 (14.2)	24.0 (2.7)
1-dotriacontanol (C_32_H_66_O)	78.0 (7.9)	2.0 (0.2)
1-tetratriacontanol (C_34_H_70_O)	20.0 (2.0)	0.0 (0.0)
Total amount	982.0 (100)	902.0 (100)

CNIC: National Center for Scientific Research, Havana, Cuba; BOC Sciences: Best of Chemical Sciences, Shirley, NY, USA.

## Data Availability

The data used to support the findings of this study are available from the corresponding author upon reasonable request.

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
