# Peer review of "Comparison of the In Vivo Efficacy of Cuban (Raydel®) and Chinese (BOC Science) Policosanol in Alleviating Dyslipidemia and Inflammation via Safeguarding Major Organs and Reproductive Health in Hyperlipidemic Zebrafish: A Twelve-Week Consumption Study"

_pharmaceuticals, 2024, doi:10.3390/ph17081103_

Round 1

Reviewer 1 Report

Comments and Suggestions for Authors

This study compares the in vivo efficacy of two different preparations of policosanol from sugarcane wax (Cuban Raydel ® vs. Chinese BOC Science) in alleviating dyslipidemia and in protecting major organs of zebrafish consuming a high cholesterol diet. The authors conclude that the Cuban product is superior to the Chinese one and attribute this difference to the different composition of the two preparations in long-chain aliphatic alcohols (LCAAs). In order to reach their conclusion, the authors have run a number of tests (plasma lipid profile and hepatic function biomarkers, liver and kidney histology, ROS production, apoptosis and cellular senescence, and ovary/testicular tissue histology and reproductive ability evaluation). Their conclusion is supported by most of their results, with the exception of the hepatic function biomarkers ALT and AST which were indeed depressed to the level of the normal diet group (ND) but did not differ between the groups receiving either product, and body weight, where no effect was recorded whatsoever.

My comments are the following:

Given the significance attributed to the different composition of the two products (rightfully so), I strongly recommend that Table S1 is moved from supplementary material to the main text (and also, correct dotriacotanol and tetratriacotanol to dotriacontanol and tetratriacontanol, respectively). Do other constituents of the two preparations differ significantly? If yes, they should be available to the reader. Last but not least, nutritional supplements are notorious for their non-consistent composition with respect to their active components. Have the authors tested the two compositions themselves or did they simply rely on the information provided by the manufacturers as hinted in Table S1?

In supplementary Figure S1, survivability appears much improved in the groups consuming HCD plus either policosanol product, compared to ND. On the other hand, there seems to be no difference between the ND and the HCD groups. This is somewhat unexpected and deserves some comment in the discussion section.

Sample collection methodology is not very clear. The authors (supplementary material S1) state that they collected 2 μL blood from the heart of each zebrafish and combined it with PBS-EDTA, then the centrifuged to obtain plasma, 5 μL of which was used in the biochemical assays. It is to be concluded that the authors pulled samples from within each group, but they need to be specific (and descriptive) about it.

The method chosen to indicate level of statistical significance in the various bar graphs is a little confusing and error-prone (e.g., in Fig6, * and *** refer to differences concerning DHE and ORO-stained area, and not SA-β-gal-stained area, and # and ### differences concerning AO and SA-β-gal-stained area, and not ORO-stained area). It would be better, I think, if the authors used horizontal bars to indicate what is compared with what each time.

Finally, I need to ask the authors to defend their choice of experimental model, given that most of animal experimentation concerning metabolic effects (that I know of, at least) makes use of small mammals (i.e, mice etc) which are considered to provide more translatable results (to humans, that is).

Comments on the Quality of English Language

The quality of English Language is clearly above average, but for a few slip-ups, such as, for instance: "A notably 2.1-fold decline..." in stead of the more appropriate "A notable 2.1-fold decline..." (repeated a number of times in the text). Also, in page 2, line 60, I believe the authors meant to write "statins" rather than "stains".

Author Response

Thank you for your insightful comments. Following the reviewer’s suggestion, we made point-to-point response and reflected on revision.

Please find attached doc as our response.

Reviewer 2 Report

Comments and Suggestions for Authors

In general, the manuscript entitled: Comparison of the In vivo Efficacy of Cuban (Raydel®) and Chinese (BOC Science) Policosanol in Alleviating Dyslipidemia and Inflammation via Safeguarding Major Organs and Reproductive Health in Hyperlipidemic Zebrafish: A Twelve Week Consumption Study is well written.

In this manuscript, the authors evaluated the influence of the two distinct policosanols, Raydel® (extracted from Cuban sugarcane wax) and BOC Sciences (extracted from Chinese sugarcane wax) on the blood lipid profile and functionality of liver, kidney and reproductive organ of hyperlipidemic zebrafish. Their results demonstrated a noteworthy impact of the policosanols in preventing high cholesterol diet-induced dyslipidemia by decreasing the plasma's total cholesterol and triglycerides level. However, compared to the BOC Sciences, Raydel® policosanol exhibited a significantly higher efficacy in reducing HCD-induced TC and TG levels. A substantial effect exclusively by the Raydel® policosanol was observed in mitigating the HCD-impaired low-density lipoprotein cholesterol (LDL-C) and high-density lipoprotein cholesterol (HDL-C) levels. The hepatic histology and immunohistochemistry (IHC) analysis revealed the higher efficacy of Raydel® over BOC Sciences policosanol to prevent HCD-provoked fatty liver changes, cellular senescence, oxidative stress, and interleukin (IL)-6 production. Besides, a significantly higher effect of Raydel® over BOC Sciences policosanol was observed on the protection of kidney, testis, and ovary morphology hampered by HCD consumption. Besides, Raydel® policosanol exhibited a notably stronger effect on the egg-laying ability of the zebrafish compared to policosanol from BOC Sciences. Furthermore, Raydel® policosanol plays a crucial role in improving embryo viability and mitigating developmental defects caused by the intake of HCD. They showed that Raydel® policosanol displayed a substantially higher efficacy over BOC Sciences policosanol to revert HCD-induced dyslipidemia, the functionality of the vital organs, and the reproductive health of zebrafish.

The manuscript has a logical structure and all conducted experiments presented within are discussed.

All research activities were performed in detail.

Statistical processing of the obtained results and their categorization were also performed systematically.

The material and methods are very well written.

The results section and discussion are very well written with all the necessary and detailed explanations.

All tables are concise, well organized and with all necessary data.

Each image is prepared with the best quality and a high resolution. 

All figures are well prepared so that all the stated and explained details are visible.

The references used are carefully selected.

English is good enough (Please check spelling and grammar once again).

The subject itself is interesting for the audience of the journal.

 I have some minor comments for improvement:

1. On page 11, lines 356 and 357 the authors need to put highlighted words in Normal.

2. On page 12, lines 373 and 374  the authors need to put highlighted words in Normal.

Besides, the reviewer marked these technical errors directly in the PDF file (Please see attached PDF file).

Author Response

(The authors gave the same response as above.)
